# Pathways of Adolescent Life Satisfaction Association with Family Support, Structure and Affluence: A Cross-National Comparative Analysis

**DOI:** 10.3390/medicina58070970

**Published:** 2022-07-21

**Authors:** Apolinaras Zaborskis, Aistė Kavaliauskienė, Elitsa Dimitrova, Charli Eriksson

**Affiliations:** 1Faculty of Public Health, Medical Academy, Lithuanian University of Health Sciences, LT-44307 Kaunas, Lithuania; 2Faculty of Odontology, Medical Academy, Lithuanian University of Health Sciences, LT-44307 Kaunas, Lithuania; aiste.kavaliauskiene@lsmuni.lt; 3Institute for Population and Human Studies, Bulgarian Academy of Sciences & Plovdiv University Paisii Hilendarski, 1000 Sofia, Bulgaria; elitsa_kdimitrova@yahoo.com; 4Department of Public Health, Stockholm University, SE-10691 Stockholm, Sweden; charli.eriksson@su.se; 5Department of Learning, Informatics, Management and Ethics, Karolinska Institute, SE-17177 Stockholm, Sweden

**Keywords:** adolescents, life satisfaction, family support, family structure, family affluence, path analysis, cross-national comparison, HBSC study

## Abstract

*Background and Objectives*: Despite the importance of life satisfaction for health and well-being, there is a paucity of cross-national comparative studies in life satisfaction related to the family environment. The present research examined the pathways of life satisfaction association with perceived family support and other family environment variables among adolescents aged from 11 to 15 years in 45 countries. *Materials and Methods*: Samples from the Health Behaviour in School-aged Children (HBSC) survey in 2017/2018 were analysed (*n* = 188,619). Path analysis was applied to evaluate the associations among the study variables. *Results*: A positive association between the life satisfaction score and high family support was identified in all 45 countries (standardized regression weight ranged from 0.067 to 0.420, *p* < 0.05). In majority of countries, living with both parents and higher levels of family affluence had a positive effect on adolescent life satisfaction both directly and indirectly through family support. In the described path model, the proportion of life satisfaction score variance that was accounted for by family support, family structure, family affluence, gender and age was up to 25.3%. The path models made it possible to group the participating countries into two clusters. In the first cluster (10 countries) the Eastern and Southern European countries dominated, while the second cluster (35 countries) united the countries of Western and Central Europe. *Conclusions*: There is evidence that countries with high level of adolescent life satisfaction differ in the high rate of intact family structure and the strong relation between family support and perceived life satisfaction.

## 1. Introduction

Life satisfaction is an assessment of overall well-being and a key predictor of many life outcomes. For example, better life satisfaction may act as a buffer against the negative effects of stress [1,2] and the development of wide range of psychopathological behaviours, such as violent behaviour or substance use [3,4,5], suicidal and self-harm behaviour [6,7]. Research shows that life satisfaction is associated with self-rated health, physical and mental health, and health-related quality of life [8,9,10]. In particular, low life satisfaction is associated with negative self-assessment of health as well as higher aggression and bullying [11,12,13]. It was also demonstrated that the adolescents who consider themselves to be more physically attractive report higher life satisfaction [14], they feel also more socially competent [15], have more friends on social media [16], have higher self-esteem [17], and experience more positive emotions [18].

Studies show that happiness emphasises the emotional experience, while life satisfaction includes also a cognitive component [19]. Myers and Diener [20] studied happiness and life satisfaction as components of subjective wellbeing. Subjective wellbeing is defined by these authors as a presence of a positive affect, absence of a negative affect and life satisfaction, i.e., the subjective wellbeing is life satisfaction, frequently experienced positive emotions (joy, affectivity) and rarely experienced negative emotions (sorrow and anxiety). Many researchers consider these three elements synonymous (e.g., [21,22]). Research shows that life satisfaction can be viewed as an important strength that facilitates young people’s development [23]. Life satisfaction is considered a hallmark of excellent mental and physical health and resilience throughout the course of life [24].

On the other hand, determinants of adolescent life satisfaction were also studied and many of them have been identified. The results from the Health Behaviour in School-aged Children (HBSC) surveys showed significant cross-national variation in the level of life satisfaction among adolescents [24,25]. However, despite the diversity of results, it has been revealed that life satisfaction decreases significantly during transitions in adolescence from 11-year-olds to 15-year-olds and is lower among girls [26]. The impact of social and material resources on life satisfaction of adolescents have been in focus of several research studies, both national [27,28,29] and cross-national [26,30,31,32,33,34,35]. It has been shown that adolescent life satisfaction is strongly influenced by life experiences and relationships [17,36,37,38,39]. Numerous studies also indicate that adolescent life satisfaction is associated with a range of different family characteristics [27,37,40,41,42,43,44,45].

Family structure is an important and apparent family characteristic. Defined by the parents and/or other persons with whom children live, family structures can be considered either intact or non-intact. Intact/nuclear families consist of children living with both their mother and their father, while non-intact/broken families can consist of single-parent families or stepparent families, usually because of parental divorce, or other non-parent adults. An intact family has been shown to be indicative of better health outcomes during adolescence as compared with a non-intact family [40,43]. In the case of parental divorce or living in a family without both biological parents the family cannot provide full support for the adolescent, which leads to less life satisfaction [23,46]. Studies have also found that children living with a single parent are much more likely to live in poverty than children living with both parents [47,48]. Thus, families of divorced parents often experience lowered affluence, which can lead to the inability to obtain basic needs including adequate amounts of food, clothing, educational support and, consequently, decrease children’s quality of life and well-being [10,28,49]. Family affluence and easy communication with parents was also noted as very important for adolescents’ life satisfaction [50]. Non-intact family structure may also negatively affect children’s life satisfaction in direct way, e.g., children who have experienced parental separation may suffer from emotional distress and loss of regular contact with the non-residential parent [41,43,46,51]. Although these results are consistent with a large prior body of epidemiological research, there have been some studies that have shown inconsistent associations [23,46].

Research shows that family/parental support also is linked to young people’s life satisfaction and well-being [52]. However, studies on the relationship between life satisfaction and family support during adolescence are relatively recent and there are still gaps in the literature in this respect [36,45]. A possible explanation for the delay in the development of research in adolescent life satisfaction study area could be related to the lack of consensus about the operationalisation of the family support construct and its conceptual relationships with other family variables [37].

Family support can be defined as positive parent–child interactions grounded in open communication and high parental sensitivity and responsiveness to the child’s needs [53]. It is relevant to adolescent health and healthy development in two aspects, from a beneficial impact on the psychological well-being of adolescents’ perspective [36,43,45], as well as from a protecting against poor mental health outcomes [54,55] and health-risk behaviour [56,57,58,59] perspective. Love, support, trust, and optimism from their family make adolescents feel safe and secure, and are powerful weapons against peer pressure, life’s challenges, and disappointments [60]. Supportive communication with parents seems to buffer the negative effects of electronic media communication [16]. However, the role of parents and parent–adolescent relations undergo a process of change through the maturation of the child during adolescence and subsequent transition from childhood to adulthood. Even though family support decreases from early to late adolescence [24] (pp. 31–34), parents continue to play a fundamental role in adolescent development, socialisation, health and well-being, and this role may be as important as it is in the early developmental stages, even if it is different and less noticeable [61].

A sizeable body of literature cited above demonstrates that perceived family support and living with both parents are linked to positive psychological outcomes, including high levels of well-being and life satisfaction in adolescence. One limitation of the literature is that it tells us little about the interaction between family support and family structure, especially if the socio-economic environment of the family, the age and gender of the subjects are considered. Thus, it must be in the interest of the next generation of research to describe the relevant pathways of the relationship, including possible biological and behavioral mechanisms, and to explain why this relationship exists [62,63].

Another limitation of this literature is that a few cross-cultural studies have examined family/parental support of adolescents. Much of the extant research on the effects of familial factors on adolescent well-being has been limited to a single country which may differ in family practices. Cultural differences in family models may impact the ways in which parents support their adolescent children [64]. Meanwhile, several studies also suggest that the effect of parental support on adolescent well-being is relatively uniform across cultural contexts [64,65]. For instance, a cross-cultural study [64] compared the association of perceived parental support to positive self-belief and distress levels among adolescents from United States, China, South Korea, and Japan. The results suggested that although the levels of parental support differed by cultural context, the perceived parental support influenced positive self-beliefs equally across cultural groups.

Family environment and health behaviours during adolescence is one of the foci of the cross-national Health Behaviour in School-aged Children (HBSC) study which involves a wide network of researchers from more than 50 countries and regions [66]. The previous reports of the study have highlighted that over two thirds of adolescents reported high levels of support from their family but a wide cross-national variation was observed [24,25]. Moreover, the association between familial factors and adolescent life satisfaction may vary depending on the social and cultural context [24] (pp. 31–34). Thus, HBSC data provide us with an excellent basis for meeting new challenges in research of the relationship between adolescent life satisfaction and familial factors.

In line with the strengths and limitations listed above, the present article has three aims. First, we aimed to empirically test the path system of relationships, which exist between family support perceived by the adolescent, family structure, and family affluence in terms of their prediction on adolescent life satisfaction, whilst controlling for gender and age. In line with this objective, the first hypothesis was that perceived family support is related to adolescent life satisfaction by interacting with family structure and family affluence and depending on the adolescent’s gender and age. Second, the study is also aimed to investigate the variation of the chosen path model across 45 HBSC countries. Noting that the strength of the relationship between family support and adolescent life satisfaction varied across countries, a second hypothesis was formulated which claims that the relationship between a higher level of adolescent life satisfaction and higher family support is consistent across countries regardless of their socio-cultural background. The last objective of this study was to check whether the identified relationships are similar among adolescents in different groups of countries.

## 2. Materials and Methods

### 2.1. Subjects and Study Design

The data were obtained from the HBSC study, a cross-national survey with support from the World Health Organization (WHO, Europe) which was completed in 2017/2018 in 45 countries, including 43 European countries and regions (considered alone as countries, i.e., England, Scotland, and Wales), Canada, and Israel. More detailed background information about the study is provided on its website [63], in the international reports [24,25], and the study protocol [67].

The population selected for sampling included 11-, 13-, and 15-year-old adolescents. Sampling was conducted in accordance with the structure of national education systems within countries. In the majority of countries, the primary sampling unit was the school class, and students of the 5th, 7th and 9th grades were targeted. School response rates within countries were in the range from 15.6% (Germany) to 100% (Bulgaria and Kazakhstan) (median 82.9%) [68].

The data were collected by means of self-report standardised questionnaires. The surveys were administrated in school classrooms. Researchers strictly followed the standardised international research protocol to ensure consistency in survey instruments, data collection, and processing procedures [67,69]. Student response rates within participating classes varied between 53.7% (Germany) and 98.6% (Albania) (median 83.2%) [68]. National datasets were cleaned by HBSC data managers and merged into the international dataset. After combining data from 45 countries, it included 228,979 records. The present analysis used records of 188,619 students who reported all variables of the model structure.

### 2.2. Ethics

The study was conformed to the principles for research outlined in the World Medical Association Declaration of Helsinki involving health promotion, safeguard, well-being, and rights of human subjects. National teams obtained ethical consent from the institutional ethics committee(s), when required. Parental consent was passive in most countries. Pupils were informed orally and in writing that participation in HBSC was voluntary. Students did not provide any personal details (such as name, classroom, teacher), making them completely anonymous and ensuring the students’ confidentiality [67].

### 2.3. Measures

Adolescent life satisfaction was the only outcome (dependent) variable of this study. It was measured by a single item and rated using a visual analogue scale known as the Cantril (1965) ladder with 11 steps: the top indicates the best possible life and the bottom the worst [70]. Respondents were asked to indicate the ladder step at which they would place their lives at present (from 0 to 10). The literature states that measuring happiness as well as life satisfaction by a single item is reliable, valid, viable in community surveys and in cross-cultural comparisons [70,71,72,73].

Family support was measured using a Family dimension (4 items) of the Multidimensional Scale of Perceived Social Support [74]. Young people were asked how they feel about the following statements: My family really tries to help me; I get the emotional help I need from my family; I can talk about problems with my family; My family is willing to help me make decisions. The respondents rated each item on a seven-point Likert-type scale, ranging from “very strongly disagree” (score of 0) through to “very strongly agree” (score of 6). The sum score was calculated as a sum of response scores to the four questions on family support ranging from 0 to 24 points (higher score corresponded to higher family support). Following previous studies [75,76], a sum score of 20 or more on MSPSS was categorised as high perceived family support. Cronbach’s alpha of the scale was 0.937.

Family structure. The family structure variable examines with whom an adolescent lives all or most of the time, including biological mother and father, stepmother (father’s partner), stepfather (mother’s partner), living in a foster or children’s home, or living with someone/somewhere else. Within the present analysis, the categories that were created comprise the groups of adolescents who live with both biological parents (intact family), and all the others (non-intact family).

Family affluence was assessed through the Family Affluence Scale (third revision), which was specially developed for the HBSC study [77]. The scale is a validated measure for the material affluence of a household based on the following six items owned by the family: number of computers, number of cars, number of bathrooms, number of travels/holidays abroad, having own bedroom, and having a dishwasher. A family affluence score (FAS) was calculated by summing the points of the responses to these six items. Higher FAS values indicated higher family affluence. In accordance with the HBSC reports [24,25], this indicator was recoded into three country-specific groups. The first group included those in the lowest 20% (reference group), the second included those in the medium 60%, and the third group included those in the highest 20% of the FAS.

Gender and age (11-, 13-, and 15-year-old) of the adolescent were also recorded during data collection.

### 2.4. Statistical Analysis

First, we used descriptive statistics to characterise the study population and describe the proportions and means of the variables of interest. Missing values, outliers, and normality were checked prior to analysis to avoid violation of statistical principles and procedures. Proportions and mean values of analyzed variables were estimated for data of each country and for aggregated data of all countries with weighting data by country sample size to ensure that the sample was representative of the general population. Comparison of the means of life satisfaction score between groups of adolescents was performed using one-way ANOVA test, while Bonferroni test was performed as a post hoc analysis for comparison of the means between three groups of adolescents. This analysis made it possible to determine the strength of the relationship between life satisfaction and other variables and the “direction” (positive or negative) of the relationship. In these as well as in subsequent analyses, a significance level was set at *p* < 0.05. Descriptive analyses were performed with SPSS (version 21.0; SPSS Inc., Chicago, IL, USA, 2012).

Then, based on logical and prior analyses, we developed the causal model to assess the pathways to adolescent life satisfaction. Hence, a structural equation model was developed. Using path analysis [78,79,80] the model examined the hypothesised causal relationships of life satisfaction with family support, family structure and family affluence adjusting data for gender and age. These relationships were assumed to be unidirectional. In this model, gender, age, and family structure were considered exogenous variables, while life satisfaction, family support, and family affluence were considered endogenous variables, so they had error components (an example of the path diagram is presented in Results). Because gender, age, and family structure were categorical variables, family supports (low/high) and family affluence (low/medium/high) were also used as categorical variables to obtain comparable regression coefficients. A structural equation modelling was conducted to assess the final model using an unweighted least squares estimation method, given its applicability to the categorical nature of data [79,81]. The model provided standardised regression coefficients (β) showing the strength of the association between the connected variables. The indirect effect of the predictor on life satisfaction was calculated as the product of the regression coefficients found in its path. Squared multiple correlations (R^2^) were displayed for each endogenous variable which is the proportion of variable variance that is accounted for by its predictors. The χ^2^/(degree of freedom df) statistic was used to assess the magnitude of the discrepancy between the sample and fitted covariance matrix, where *p* > 0.05 indicated that the model and data were consistent. Because this statistic is sensitive to sample size, model fit was also evaluated using the root mean square error of approximation (RSMEA) and other three goodness-of-fit statistics that are widely applied: the incremental fit index (IFI); the Tucker–Lewis index (TLI); the comparative fit index (CFI). Note that the RMSEA statistic measures how far our model is from a perfect model, while, on the contrary, IFI, TLI and CFI compare the fit of a hypothesised model with that of a baseline model (i.e., a model with the worst fit) [81]. RSMEA value lower than 0.09, and IFI, TLI, CFI values higher than 0.9 indicated good model fit to real data [80,81]. Path analysis was performed using AMOS 21 (SPSS Inc., Chicago, IL, USA, 2012) [75]. Finally, a hierarchical cluster analysis was performed to classify the countries by the main characteristics of the model.

## 3. Results

### 3.1. Descriptive Results

Sample size and socio-demographic statistics are shown in Table 1. Analyses were based on responses of 188,619 students. Of them, 48.1% were boys and 51.9% were girls. The country samples were evenly distributed between gender groups, except for Albania and Greenland, for which the difference between the proportions of boys and girls was more than 10 percentage points. Across countries, the age groups of 11, 13, and 15 years achieved nearly equal proportions, which were maintained throughout the whole sample (31.7%, 33.9%, and 32.4%, respectively). The proportions of subjects in “low”, “medium”, and “high” family affluence groups were relative for each country; across countries, they varied around 20%, 60%, and 20% groups, respectively. Family structure distribution showed wide variation across countries, ranging from 47.9% of adolescents living with both parents in Azerbaijan to 91.5% in Albania (72.9% in the total sample).

Descriptive statistics of life satisfaction score and family support are shown in Table 2. Among the 45 countries involved in the analysis, the mean of life satisfaction score was ranging between 7.29 (Canada) and 8.61 (Kazakhstan), while in the whole sample it was equal to 7.82 (95% CI: 7.81–7.83). The lowest prevalence of high family support in the study was found in Bulgaria (38.6%), while the highest proportion of high family support was found in North Macedonia (89.9%), and the weighted prevalence of high family support in the whole sample was 71.3% (95% CI: 71.2–71.4).

At the country level (using data from Table 2), a positive correlation was observed between the mean of life satisfaction score and the proportion of high family support (r = 0.429; *p* = 0.003). A highly significant association between adolescent life satisfaction and family support was confirmed in all three age groups, although the mean of life satisfaction score decreased significantly with age regardless of the level of family support (Table 3).

The possible association of life satisfaction with other selected factors was also investigated. The summary results of the analysis of aggregated data presented in Table 4 show that girls were less likely than boys to feel satisfied in life, but the gender difference was small; it was insignificant among 11-year-olds. Analogously, data presented in Table 5 show that living with both parents and family affluence were both positively related to adolescent life satisfaction.

### 3.2. Results of Path Model Analysis

The hypothesised model was tested in the AMOS program.

Table 6 presents the standardised regression weights of the path model estimated from aggregated data of all 45 HBSC countries as well as the assessments of these characteristics obtained from the path analysis by countries. It can be noticed that family support has a positive effect on adolescent life satisfaction, i.e., higher family support increases the chance of a higher life satisfaction score. This effect was the largest of all the associations in the model (in analysis of aggregated data β = 0.286). By countries, the effect as measured by the standardised regression weight ranged from 0.067 (Georgia) to 0.420 (Estonia) with a median of 0.310 and was significant in all HBSC countries.

Analysing the aggregated data from all countries, the model shows that girls compared to boys had a little lower life satisfaction score and were also likely to have reduced family support, however, data analysis by countries has shown that these relationships are diverse. In contrast, age had a significantly greater than gender effect on both life satisfaction and family support. Across the majority of countries, it was observed that older adolescents felt less life satisfied and experienced less family support. It should be noted that age had both a direct effect on life satisfaction and an indirect effect on life satisfaction through family support. When data from all countries were analyzed, the latter component was equal −0.121 × 0.286 = −0.035, which shows that the effect of family support on adolescent life satisfaction was additionally reduced due to older age.

The family structure also plays an important role in this model. Across most countries, not living with both biological parents had a direct negative effect on adolescent life satisfaction. This predictor also had a statistically significant negative effect on family support (in data of all countries β = −0.103), so its indirect negative effect on life satisfaction through family support can also be predicted. Another indirect effect of broken family structure on adolescent life satisfaction may have occurred through FAS, which was adversely affected by family structure in majority of countries. In turn, across most countries, FAS had a positive direct effect on life satisfaction: adolescents from more affluent families felt more satisfied with their life (in data of all countries β = 0.089). FAS also had a positive effect on family support, and, thus, indirectly increased adolescent life satisfaction.

In the described path model, the proportion of life satisfaction score variance that was accounted for by family support, family structure, family affluence, gender, and age (statistic R^2^) was up to 25.3% (Estonia). In analysis of aggregated data from all countries, these predictors accounted for 14.2% of the variance of life satisfaction (see Table 6).

Figure 1 presents the described path model with standardised regression weights and variances of endogenous variables which were calculated using data from Estonia. It was chosen because of the maximum R^2^ (0.253 or 25.3%) obtained by the model. Among the countries, data of Estonian adolescents also showed the highest effect of family support on life satisfaction (β = 0.42). Considering the effects of gender, age, family structure, and family affluence on family support, the effect of the latter on life satisfaction was reduced by a total of 0.07 units. Thus, the direct effect of family support on life satisfaction can be predicted to be at the level of 0.49, which was confirmed by a separate path model without indirect effects (data are not presented).

Table 7 presents goodness-of-fit indices of the path model that was estimated in analysis of aggregated data from all 45 HBSC countries and shows the assessments of these indices when they were obtained from the data analysis by countries. The χ^2^/(df) statistic that was applied to assess the magnitude of the discrepancy between the sample and fitted covariance matrix indicated consistency (*p* > 0.05) the model and data only in a few countries due to its sensitivity to sample size. The RMSEA index showed good model fit for the aggregated data of all countries and for the data of each country. The IFI and CFI indices showed a similar result for model fit testing. The TLI index for all country data also demonstrated good model fit, but varied more widely between countries and showed good model fit for only 28 countries. Our analysis confirmed that the χ^2^/df statistic that was used to assess the magnitude of the discrepancy between the data and fitted covariance matrices is sensitive to sample size. Therefore, in a large data sample from all countries and from 32 countries with relatively large data samples, the values of this statistic were unacceptable.

### 3.3. Groups of Countries According to the Characteristics of the Path Model

As 45 countries were involved in the current round of the HBSC study, there were the same number of different path models constructed. Consequently, we tried to group the countries according to the main characteristics of the path model. Some of these model characteristics (R^2^, LS mean, β1 (LS ← FS)) were selected using the results of correlation analysis, which evaluated the relationships between R^2^, mean of life satisfaction score and path regression weights estimated from the country level data (Table 8). The analysis revealed that R^2^ had the maximal correlation (r = 0.935, *p* < 0.01) with the regression weight of family support on life satisfaction score but was uncorrelated with the regression weights of age (r = 0.076, *p* = 0.622) and family affluence (r = −0.187, *p* = 0.220) on life satisfaction score. Meanwhile, higher R^2^ values were observed in countries with lower mean of life satisfaction score (r = −0.591, *p* < 0.01). Additionally, the strength of life satisfaction association with age (β4 (LS ← AGE)) was chosen for classification of the models because it, as well as the strength of life satisfaction association with family support (β1 (LS ← FS)), was significant in all countries.

Using a hierarchical cluster analysis and the four variables just mentioned, the countries were classified into two clusters. The first cluster included 10 countries (Albania, Azerbaijan, Armenia, Croatia, Kazakhstan, North Macedonia, the Republic of Moldova, Romania, Serbia, Spain). The second cluster combined the remaining 35 countries. The data in Table 9 show that the selected model characteristics differed significantly between country clusters. In contrast to the second cluster, countries in the first cluster differed in higher levels of life satisfaction but lower R^2^, weaker positive effects of family support on life satisfaction, and more negative direct effects of age on life satisfaction.

## 4. Discussion

In line with the findings from literature review and results from descriptive analysis of empirical data of the survey among 11–15-year-old adolescents from the 45 HBSC countries in 2017/2018 we designed a path model to explore the relationships between life satisfaction, perceived family support, family structure and family affluence including adjustment for adolescent’s gender and age. Using this model, two hypotheses were tested in the current study, and results of the analysis confirmed both hypotheses. First, for the majority of countries, it was found that perceived family support is related to adolescent life satisfaction by interacting with family structure and family affluence and depending on the adolescent’s gender and age. Second, it was confirmed that a positive association between adolescent life satisfaction and perceived family support is consistent across all countries despite their socio-cultural background. These findings are extending the current understanding of the potential mechanisms that might explain how family support, family structure and family affluence can influence adolescent life satisfaction from the cross-national study perspective.

Data analysis began with descriptive statistics which confirmed the facts about adolescent lifestyle and family environment described in the recent and previous rounds of the HBSC study [24,25]. Despite the limited originality of these results, they were helpful to set up the path system of relationships between variables and to control model parameters. Indices of the goodness-of-fit showed good path model fit for the empirical data showed perfect path model fit for the aggregated data of all countries and for the data of each country.

In this study we measured life satisfaction with a single item answered on a 0–10 score scale. Adolescents aged 11–15 years were inclined to rate their life satisfaction with a high score (7.8 out of 10), the prevalence of which across countries was similar but declined between ages 11 and 15, especially among girls [25]. There was no gender difference in the mean of life satisfaction score among 11-year-olds, while at ages 13 and 15, boys reported higher life satisfaction scores than girls. These findings were supported by the results from path analysis which indicate a consistent direct relation of life satisfaction to age and gender of adolescents across all countries. The path model in addition shows that older adolescent age and female gender negatively influence family support and thus throughout this factor have an indirect negative effect on life satisfaction, i.e., older adolescents as well as girls perceive less family support and, therefore, have lower life satisfaction. Age and gender differences in life satisfaction may be perceived as a result of the physical, psychological and social changes experienced in adolescence (hormonal changes, changes in the body, thinking, social relationships, roles, identity search), as well as gender characteristics (boys tend to experience negative emotions less frequently than girls, they cope with their emotions better, they report more satisfaction with their appearance and are more engaged in physical activities) [36,49,82,83]. Similarly, older adolescents report having less family support compared to their younger counterparts due to the beginning of the individualisation process where the relationships with family members move from asymmetrical to a more symmetrical interaction, with the adolescents treated as more autonomous individuals [84]. According to our findings, girls reported having a lower level of family support compared to boys, which is in line with other research that found that girls, more often than boys, start experiencing problems of connectedness with parents in early adolescence, while boys have a higher need of psychological separation and independence from parents [49].

In this study, we tried to determine the pathways of how perceived family support, family structure, and family affluence are related with adolescent life satisfaction. Several pathways of the studied system of relationships between familial variables and children and adolescent life satisfaction/well-being were reported in previous research, namely, effect of family socio-economic status [24,25,27,29], family structure [41,43,46,51], and family/parental support [36,37,42,44,45]. The negative impact of broken family structure on family affluence was also described [10,28,47,48,49]. For the most part, the results that followed from our descriptive and path analysis were in line with the findings of the studies presented. However, through the path analytic method we were able to combine all these pathways into one system of relationships and explore it from an integrated point of view. We found that family support had a positive direct effect on life satisfaction, however, its strength could be diminished due to broken family structure and/or low family affluence. These observations confirmed the first hypothesis that perceived family support is related to adolescent life satisfaction by interacting with family structure and family affluence and depending on the adolescent’s gender and age. It was also found that the positive direct effect of family support was maintained among all countries. This finding confirmed the second hypothesis on consistency of the positive relationship between family support and adolescent life satisfaction.

The findings of our study allow us to conclude that the path system of relationships, which exist between family support, family structure, and family affluence controlling for gender and age, is an important asset in helping to predict the level of adolescent life satisfaction. The statistic R^2^, which reflects the proportion of the outcome variable variance accounted for by its predictors, was up to 25%. According to Cohen’s effect size benchmark, an R^2^ higher than 25% is regarded as high [85]. Such a value can be considered high compared to other models that attempt to predict human behaviour [86]. In the presented model, R^2^ had a strong positive relationship with the strength of association between family support and life satisfaction, but was negatively associated with the level of life satisfaction. The calculated correlations allowed us to identify the conditions on which the quality of the model depends.

This study is unique as a series of path models was constructed using data from a variety of countries. Despite the fact that each model had unique characteristics, some similarities within several model groups could be seen, so cluster analysis was used to classify the countries according to the model characteristics. For this purpose, we selected four model characteristics which were considered consistent from a statistical point of view. These were R^2^, mean level of life satisfaction, the strength of life satisfaction association with family support, and the strength of life satisfaction association with age. According to these characteristics of the path model, the countries were classified in two clusters. The first cluster was dominated by Eastern and Southern European countries, while the second cluster mainly united the countries of Western and Central Europe. Notably, adolescents from the countries of the first cluster in comparison with the countries of the second cluster tend to have higher rates of life satisfaction; however, they showed lower levels of absolute FAS [87]. By the macroeconomic indices (e.g., gross national income per capita) the countries of the first cluster can be also characterised as less developed and poorer countries [88]. These results may appear to be contradicted by the statement that adolescents from more affluent families reported higher levels of life satisfaction across all countries, as revealed in this and previous rounds of the HBSC study [24,25]. This fact was also confirmed by the path model, which found a positive direct dependence of life satisfaction on FAS level 42 out of 45 countries. In looking for reasons to explain this fact, it should first be noted that the countries in the first cluster had a significantly higher percentage of intact families [25]. Second, what appears to be the most important, a significantly stronger relationship between family support and adolescent life satisfaction has been found in these countries. Thus, despite the economically poorer family life that exists in these countries, a lower rate of damaged family structures and stronger relationship between family support and life satisfaction ensure higher adolescent life satisfaction.

High life satisfaction is important for the general quality of life of adolescents and for mental health in general [1,2,3,4,5,6,7,31,32,33,34,35,36]. It is therefore a public health and social policy concern of the greatest interest to ensure high life satisfaction for all adolescents, taking into account their health needs according to age and gender. In line with previous research [40,41,42,43,44,45,46,47,48,49,50], this study showed that perceived family support is related to adolescent life satisfaction. Moreover, this relation is intermediated with family structure and family affluence. Therefore, each country should review local policies to determine the extent to which they address the social determinants of the family, the provision of services towards the creation of structural changes in the family wealth, and in the strengthening of the family institution so that children could grow up in family with both parents, in particular. On the other hand, prevention and intervention programs that aim to enhance the well-being of adolescents may be improved by the inclusion of strategies to help parents to recognise and increase the understanding of the importance of family support for a child’s emotional development, happiness, and life satisfaction [24,25].

### Strengths and Limitations

The present study was strengthened using a large and culturally diverse sample of adolescents; therefore, the performance of the path model could be examined immediately in 45 samples and generalised across countries. Uniform methodology of the HBSC study ensured the comparability of data between countries. The findings of the study increased our knowledge of adolescent well-being in the context of family functioning and contributed to the existing literature in this field. Our study was not exhaustive; rather, we intended to give a sense of the importance of path analysis for the understanding of the system of relationships, which exist between perceived family support, family structure, and family affluence in terms of their prediction role in adolescent life satisfaction.

We also hereby acknowledge two kinds of limitations of the present study.

The limitations of the first kind are related to the survey methodology. This study is cross-national. Although research centres in each country have been required to adhere strictly to the research protocol [64], there is still room for methodological differences between countries in various aspects of data collection (e.g., imprecise translation of questions into national language, different organisation of data collection), and some of these might affect the value of relationship between measures. This study relies only on adolescents’ self-reported data and is thus a subject to potential response bias. For example, using sensitive questions, such as questions about relations with parents, can be affected by the possibility for social fear bias in adolescent responses. Every effort was made to minimise that possibility by ensuring strict anonymity of respondents. The main concern was the adolescent life satisfaction assessment, which was constructed by a respondents’ self-reported single-item scale. Such an approach was recognised by the scientific research committee of the HBSC study to be reliable, valid, and viable to measure adolescent life satisfaction and has been already implemented in several rounds of the study. A single-item measuring of life satisfaction/happiness was also confirmed by other studies [71,72,73,89]. The Cantril ladder has been recommended for use as a valid and reliable measure of life satisfaction across nations [72].

The limitations of the second kind are related to the data processing and analysis of the relationship between variables, family support, and life satisfaction, in particular. Although family support was built on four-item scale [74], we assessed only its summed score, and not, for instance, the weight of each item on the unobserved variable named “family support”. This solution was chosen in order to have a simpler model and an easier interpretation of the results. For this reason, other variables that were used in the HBSC study, such as adolescent communication with mother and father, or support from friends, were not included in the model. While data are available from many countries, the most appropriate approach for the analysis of associations between individual variables in the cross-cultural context might be multilevel modelling, such as those demonstrated in other studies [49]. In the present study, we did not use multilevel modelling, because the study aimed to identify the relevant structure of relationships between adolescent life satisfaction and main familial characteristics. Analysis of the correlation between adolescent life satisfaction and country level predictors was still assigned for testing of external validity of the path model. Finally, the analysis of cross-sectional questionnaire data could only suggest associations among variables, but not causation [90]. However, the path model allowed predicting the directionality of associations because the model’s goodness-of-fit statistics satisfied a given level [78].

## 5. Conclusions

The findings of this study extend the current understanding of the potential mechanisms of relationships which exist between perceived family support, family structure, and family affluence in terms of their prediction on adolescent life satisfaction, whilst controlling for gender and age. The path model developed in the study underlines the importance of perceived family support on adolescent life satisfaction which is consistent across countries regardless of their socio-cultural background. Family structure and family affluence in conjunction with gender and age are also important factors which can predict adolescent life satisfaction both directly and indirectly through the family support. Countries with a higher level of adolescent life satisfaction differed by a high rate of intact family structure and strong relation between family support and perceived life satisfaction. Prevention and intervention programs that aim to enhance the well-being of adolescents may be improved by the inclusion of strategies to help adolescents achieve and maintain high-quality family support and grow up with both parents in a wealthy enough family.

## Figures and Tables

**Figure 1 medicina-58-00970-f001:**
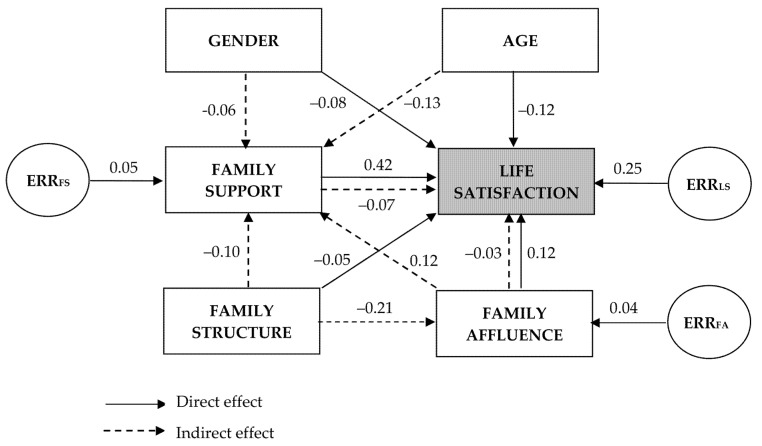
Path model and structural pathways to adolescent life satisfaction: a case of data from Estonia. Notes: the numbers on pathways are standardised coefficients; ERR_LS_, ERR_FA_, and ERR_FS_ are errors of corresponding variables which represent unmeasured variables.

**Table 1 medicina-58-00970-t001:** Sample size and socio-demographic characteristics of study participants, by countries.

Country	Sample Size	% of Boys	Age Groups (%)			Family Affluence (%)			% of Intact Families
			11 Years	13 Years	15 Years	Low	Medium	High	
Albania	1440	42.6	39.2	15.6	45.2	20.1	60.2	19.7	91.5
Azerbaijan	3971	47.5	34.8	33.2	32.1	20.4	58.9	20.7	47.9
Austria	3514	48.6	30.4	35.6	34.0	17.5	62.2	20.2	71.0
Armenia	3349	46.3	30.8	35.3	33.9	22.4	58.1	19.5	90.0
Belgium (Flemish)	3696	48.7	38.5	28.4	33.2	16.8	61.6	21.6	69.0
Belgium (French)	3204	47.6	54.4	22.5	23.0	18.3	60.9	20.8	68.3
Bulgaria	4173	48.5	36.3	29.9	33.8	22.3	60.1	17.6	76.6
Canada	8536	46.9	24.5	39.7	35.8	19.6	57.7	22.7	70.5
Croatia	4139	48.9	29.2	29.5	41.3	19.6	57.9	22.5	83.3
Czech Republic	10101	50.0	31.5	34.6	33.9	19.8	62.2	18.0	70.0
Denmark	2776	47.6	40.5	34.5	25.0	22.6	61.5	16.0	73.8
Estonia	4330	49.7	32.7	34.2	33.1	21.1	59.6	19.3	67.4
Finland	2830	48.9	30.4	35.4	34.2	21.9	59.7	18.3	73.7
France	7227	48.6	33.4	40.3	26.3	16.0	66.3	17.7	68.5
Georgia	3575	48.7	32.3	35.6	32.2	22.3	60.8	16.9	84.5
Germany	3635	45.9	29.3	33.0	37.7	16.8	65.9	17.3	73.1
Greece	3483	49.1	31.7	34.0	34.3	14.2	64.4	21.4	81.3
Greenland	716	45.0	34.9	39.7	25.4	15.6	61.9	22.5	54.8
Hungary	3351	46.1	32.7	36.5	30.8	18.9	62.3	18.8	70.2
Iceland	6384	49.5	32.3	36.0	31.7	16.1	60.0	24.0	70.0
Ireland	3124	49.5	33.9	36.9	29.2	20.4	63.2	16.4	76.3
Italy	3737	47.2	33.4	35.0	31.6	15.9	63.2	20.9	79.7
Kazakhstan	3935	49.1	35.2	32.9	31.9	21.5	60.5	18.0	70.7
Latvia	3953	48.8	34.3	34.5	31.2	17.9	62.9	19.2	62.3
Lithuania	3462	49.7	34.9	33.6	31.4	17.4	62.7	19.8	68.0
Luxembourg	3285	49.1	32.0	32.3	35.7	21.1	61.9	17.0	67.8
Malta	2026	46.3	43.7	33.3	23.0	15.4	65.6	18.9	78.1
Republic of Moldova	4049	49.2	33.0	33.1	33.9	18.6	61.8	19.6	71.6
Netherlands	4305	48.4	30.9	36.2	32.9	20.4	58.5	21.0	77.6
Norway	2319	48.3	51.8	25.8	22.4	19.8	62.1	18.1	71.4
Poland	4689	48.4	32.2	33.2	34.6	16.5	63.7	19.8	78.2
Portugal	5231	47.8	36.3	39.2	24.5	20.3	63.4	16.2	71.1
Romania	3799	48.5	33.1	33.4	33.5	18.2	61.0	20.8	61.7
Russia	3736	47.6	25.5	31.5	43.0	21.7	59.6	18.7	68.7
Serbia	3377	47.4	26.0	30.9	43.1	18.3	60.3	21.4	76.2
Slovakia	2932	48.5	30.2	40.6	29.2	18.9	60.2	20.9	76.5
Slovenia	5175	50.0	34.3	35.7	30.0	18.6	64.5	16.9	80.7
Spain	4006	48.1	27.5	36.6	35.9	17.6	62.4	20.0	78.7
Sweden	3511	48.6	27.1	33.3	39.7	14.9	67.4	17.7	72.1
Switzerland	6451	49.5	32.0	35.4	32.6	23.0	56.1	20.8	78.3
Ukraine	5427	46.2	32.4	36.0	31.6	17.0	63.0	20.0	72.3
North Macedonia	3900	48.1	33.1	34.8	32.1	18.3	61.2	20.6	89.1
England	2664	52.4	38.5	35.5	26.0	16.3	66.6	17.1	67.5
Scotland	4205	46.7	36.7	33.8	29.5	17.6	66.8	15.6	65.1
Wales	10891	48.4	30.7	38.9	30.5	22.1	59.5	18.5	66.0
All countries ^1^	188619	48.1	33.7	33.9	32.4	18.9	61.8	19.3	72.9

Note: ^1^ Data were weighted by country sample size.

**Table 2 medicina-58-00970-t002:** Descriptive statistics of life satisfaction score and high family support, by countries.

Country	Life Satisfaction Score			% of High Family Support		
	Mean	95% CI		Proportion	95% CI	
		Lower	Upper		Lower	Upper
Albania	8.15	8.09	8.21	87.8	86.0	89.6
Azerbaijan	8.40	8.34	8.46	69.3	67.7	70.6
Austria	7.73	7.67	7.79	75.0	73.5	76.4
Armenia	8.37	8.32	8.42	77.9	76.4	79.2
Belgium (Flemish)	7.83	7.79	7.87	73.9	72.4	75.4
Belgium (French)	7.72	7.66	7.77	73.9	72.3	75.5
Bulgaria	7.84	7.78	7.90	38.6	37.1	40.0
Canada	7.29	7.23	7.35	49.7	48.6	50.8
Croatia	8.13	8.08	8.19	79.5	78.3	80.6
Czech Republic	7.81	7.76	7.86	58.8	57.8	59.7
Denmark	7.68	7.63	7.73	82.5	81.1	84.0
Estonia	7.75	7.70	7.81	75.6	74.4	76.9
Finland	7.79	7.73	7.84	68.9	67.3	70.6
France	7.68	7.62	7.73	71.8	70.7	72.8
Georgia	7.99	7.93	8.05	66.0	64.4	67.6
Germany	7.69	7.64	7.74	72.8	71.3	74.2
Greece	7.56	7.50	7.62	77.3	75.9	78.7
Greenland	7.88	7.81	7.95	52.9	49.2	56.4
Hungary	7.61	7.56	7.67	82.4	81.0	83.7
Iceland	7.64	7.58	7.70	71.1	70.0	72.2
Ireland	7.56	7.50	7.62	62.3	60.5	64.1
Italy	7.61	7.55	7.66	75.7	74.3	76.9
Kazakhstan	8.61	8.55	8.66	80.7	79.5	81.9
Latvia	7.42	7.37	7.48	66.3	64.8	67.8
Lithuania	7.93	7.87	7.99	72.3	70.8	73.8
Luxembourg	7.70	7.64	7.75	72.4	70.9	73.9
Malta	7.41	7.34	7.47	74.2	72.4	76.1
Republic of Moldova	8.24	8.19	8.29	75.2	73.9	76.5
Netherlands	7.79	7.75	7.84	81.3	80.1	82.4
Norway	7.93	7.88	7.99	83.1	81.5	84.6
Poland	7.49	7.43	7.55	59.6	58.3	61.0
Portugal	7.77	7.71	7.82	80.4	79.3	81.5
Romania	8.38	8.33	8.43	81.6	80.4	82.8
Russia	7.40	7.34	7.46	65.3	63.8	66.8
Serbia	8.25	8.20	8.31	84.9	83.7	86.1
Slovakia	7.61	7.55	7.66	70.5	68.8	72.2
Slovenia	7.99	7.94	8.04	61.2	59.9	62.5
Spain	8.12	8.07	8.17	79.7	78.4	80.8
Sweden	7.50	7.44	7.56	79.1	77.8	80.4
Switzerland	7.68	7.63	7.74	75.8	74.7	76.8
Ukraine	7.68	7.62	7.74	57.7	56.4	58.9
North Macedonia	8.44	8.38	8.50	89.9	89.0	90.9
England	7.48	7.42	7.53	53.0	51.1	54.8
Scotland	7.64	7.59	7.70	59.5	58.1	61.1
Wales	7.64	7.58	7.69	61.9	61.1	62.9
All countries ^1^	7.82	7.81	7.83	71.3	71.2	71.4

Note: ^1^ Data were weighted by country sample size.

**Table 3 medicina-58-00970-t003:** Summary data on life satisfaction association with family support and age in adolescents from all 45 countries.

Age	Life Satisfaction Score ^1^						*p*
	Low Family Support		High Family Support		Total		
	Mean ^2^	(95% CI)	Mean ^2^	(95% CI)	Mean ^2^	(95% CI)	
11 years	7.42 ^a^	(7.38–7.45)	8.50 ^a^	(8.49–8.52)	8.27 ^a^	(8.26–8.28)	<0.001
13 years	6.85 ^b^	(6.82–6.88)	8.13 ^b^	(8.12–8.15)	7.75 ^b^	(7.74–7.77)	<0.001
15 years	6.58 ^c^	(6.55–6.61)	7.87 ^c^	(7.85–7.88)	7.42 ^c^	(7.40–7.43)	<0.001
Total	6.89	(6.87–6.90)	8.19	(8.18–8.20)	7.82	(7.81–7.83)	<0.001
*p*	<0.001		<0.001		<0.001		

Notes: ^1^ Data were weighted by country sample size; ^2^ means by age with different superscript letter are significantly different (Bonferroni post hoc test); CI—confidence interval; *p*-values were determined by one-way ANOVA test.

**Table 4 medicina-58-00970-t004:** Summary data on life satisfaction association with gender and age in adolescents from all 45 countries.

Age	Life Satisfaction Score ^1^						*p*
	Boys		Girls		Total		
	Mean ^2^	(95% CI)	Mean ^2^	(95% CI)	Mean ^2^	(95% CI)	
11 years	8.28 ^a^	(8.26–8.30)	8.26 ^a^	(8.24–8.28)	8.27 ^a^	(8.26–8.28)	0.104
13 years	7.92 ^b^	(7.90–7.94)	7.59 ^b^	(7.57–7.61)	7.75 ^b^	(7.74–7.77)	<0.001
15 years	7.64 ^c^	(7.62–7.66)	7.22 ^c^	(7.20–7.24)	7.42 ^c^	(7.40–7.43)	<0.001
Total	7.95	(7.94–7.96)	7.69	(7.68–7.71)	7.82	(7.81–7.83)	<0.001
*p*	<0.001		<0.001		<0.001		

Notes: ^1^ Data were weighted by country sample size; ^2^ means by age with different superscript letter are significantly different (Bonferroni post hoc test); CI—confidence interval; *p*-values were determined by one-way ANOVA test.

**Table 5 medicina-58-00970-t005:** Summary data on life satisfaction association with family affluence and family structure in adolescents from all 45 countries.

Family Affluence	Life Satisfaction Score ^1^						*p*
	Live with Both Parents		Don’t Live with Both Parents		Total		
	Mean ^2^	(95% CI)	Mean ^2^	(95% CI)	Mean ^2^	(95% CI)	
Low	7.68 ^a^	(7.65–7.71)	7.05 ^a^	(7.02–7.09)	7.45 ^a^	(7.42–7.47)	<0.001
Medium	7.94 ^b^	(7.93–7.96)	7.48 ^b^	(7.46–7.51)	7.83 ^b^	(7.81–7.84)	<0.001
High	8.24 ^c^	(8.22–8.26)	7.84 ^c^	(7.80–7.88)	8.16 ^c^	(8.14–8.18	<0.001
Total	7.96	(7.95–7.97)	7.43	(7.41–7.44)	7.82	(7.81–7.83)	<0.001
*p*	<0.001		<0.001		<0.001		

Notes: ^1^ Data were weighted by country sample size; ^2^ means by family affluence with different superscript letter are significantly different (Bonferroni post hoc test); CI—confidence interval; *p*-values were determined by one-way ANOVA test.

**Table 6 medicina-58-00970-t006:** Standardised regression weights (β) and squared multiple correlation (R^2^) in path analysis of the total sample of all 45 HBSC countries.

Measure	Analysis of Aggregated Data from All Countries	Data Analysis by Countries			
		Range	Median	Number of Countries with Significant Positive Effect	Number of Countries with Significant Negative Effect
β:					
β_1_ (LS ← FS)	0.286	0.067 to 0.420	0.310	45	0
β_2_ (LS ← GENDER)	−0.061	−0.057 to 0.028	−0.057	0	36
β_3_ (FS ← GENDER)	−0.022	−0.087 to 0.057	−0.028	5	22
β_4_ (LS ←AGE)	−0.142	−0.235 to –0.066	−0.141	0	45
β_5_ (FS ← AGE)	−0.121	−0.252 to 0.006	−0.133	0	42
β_6_ (LS ← FSTR)	−0.085	−0.121 to 0.029	−0.090	0	42
β_7_ (FS ← FSTR)	−0.103	−0.131 to –0.012	−0.096	0	43
β_8_ (FAS ← FSTR)	−0.121	−0.209 to 0.125	−0.127	1	40
β_9_ (LS ← FAS)	0.089	−0.015 to 0.181	0.089	42	0
β_10_ (FS ← FAS)	0.047	−0.029 to 0.139	0.049	33	0
R^2^ (%)	14.2	3.1 to 25.3	15.5		

Notes: β—standardised regression weight; FAS—family affluence score; FS—family support; FSTR—family structure; LS—life satisfaction score; R^2^ (%)—the squared multiple correlation (the variance of LS score that is accounted for by its predictors FAS, FS, FSTR, GENDER, and AGE).

**Table 7 medicina-58-00970-t007:** Summary of model fit to data of 45 HBSC countries.

Model Fit Indices	Analysis of Aggregated Data from All Countries	Data Analysis by Countries		
		Range	Median	Number of Countries with Good Model Fit
χ^2^/df	66.394	0.90 to 40.29	4.581	-
p (χ^2^/df)	<0.001	<0.001 to 0.482	<0.001	13
RMSEA	0.017	0 to 0.058	0.029	45
IFI	0.992	0.886 to 0.999	0.978	43
TLI	0.967	0.532 to 0.999	0.915	28
CFI	0.992	0.882 to 0.999	0.976	43

Notes: χ^2^/df—the minimum discrepancy, divided by its degrees of freedom; p (χ^2^/df)—*p*-value of χ^2^/df; IFI—the incremental fit index; TLI—the Tucker–Lewis index; CFI—the comparative fit index; RMSEA—the root mean square error of approximation.

**Table 8 medicina-58-00970-t008:** Correlations between R^2^, mean of life satisfaction score (LS_mean) and path regression weights estimated from the country-level data.

Country Level Statistics	Pearson Coefficients of Correlation										
	LS_mean	β1	β2	β3	β4	β5	β6	β7	β8	β9	β10
R^2^	−0.591 **	0.935 **	−0.656 **	−0.551 **	0.076	–0.602 **	–0.519 **	–0.302 *	–0.535 **	–0.187	0.556 **
LS_mean	1	−0.602 **	0.557 **	0.430 **	−0.371 *	0.488 **	0.447 **	0.005	0.637 **	0.039	–0.479 **
β1 (LS ← FS)		1	−0.673 **	−0.616 **	0.305 *	–0.569 **	–0.547 **	–0.334 *	–0.529 **	–0.359 *	0.593 **
β2 (LS ← GENDER)			1	0.522 **	−0.260	0.310 *	0.536 **	0.103	0.354 *	0.456 **	–0.534 **
β3 (FS ← GENDER)				1	−0.279	0.273	0.366 *	0.178	0.447 **	0.274	–0.274
β4 (LS ← AGE)					1	–0.077	–0.118	–0.102	–0.165	–0.118	0.189
β5 (FS ← AGE)						1	0.359 *	0.258	0.535 **	0.018	–0.209
β6 (LS ← FSTR)							1	0.225	0.557 **	0.246	–0.356 *
β7 (FS ← FSTR)								1	0.300 *	–0.007	–0.044
β8 (FAS ← FSTR)									1	–0.014	–0.274
β9 (LS ← FAS)										1	–0.216
β10 (FS ← FAS)											1

Notes: * *p* < 0.05; ** *p* < 0.01.

**Table 9 medicina-58-00970-t009:** Comparison of country clusters.

Cluster	Number of Countries	Mean (95% CI)			
		R^2^ (%)	Mean of Life Satisfaction Score	β1 (LS ← FS)	β4 (LS ← AGE)
Cluster 1	10	11.4 (7.8 to 15.0)	8.31 (8.20 to 8.42)	0.203 (0.144 to 0.261)	–0.179 (–0.199 to –0.159)
Cluster 2	35	16.4 (14.9 to 17.9)	7.68 (7.62 to 7.74)	0.301 (0.275 to 0.327)	–0.137 (–0.151 to –0.123)
*p*-value		0.004	<0.001	0.001	0.004

Note: Countries of cluster 1: Albania, Azerbaijan, Armenia, Croatia, Kazakhstan, North Macedonia, Republic of Moldova, Romania, Serbia, Spain. Countries of cluster 2: Austria, Belgium (Flemish), Belgium (French), Bulgaria, Canada, Czech Republic, Denmark, England, Estonia, Finland, France, Georgia, Germany, Greece, Greenland, Hungary, Iceland, Ireland, Italy, Latvia, Lithuania, Luxembourg, Malta, Netherlands, Norway, Poland, Portugal, Russia, Scotland, Slovakia, Slovenia, Sweden, Switzerland, Ukraine, Wales.

## Data Availability

The data presented in this study are available on reasonable request from the HBSC Data Management Centre, University of Bergen, Norway (dmc@hbsc.org).

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
