# Peer review of "Pathways of Adolescent Life Satisfaction Association with Family Support, Structure and Affluence: A Cross-National Comparative Analysis"

_medicina, 2022, doi:10.3390/medicina58070970_

Round 1
Reviewer 1 Report
First of all, thank you for the possibility to review the manuscript,
It's a good job, on a very interesting topic as it is the satisfaction with life, Family Support, Structure and Affluence in adolescents of different countries, but before being published you must incorporate the following suggestions, I ask the authors to follow them one by one,
in the introduction some important references are missing, revise it,
although it is deduced from the text, it must explain the research gap that the article tries to cover,
the explanation of the procedure carried out for the investigation needs to be improved, it is unclear,
the discussion section should be improved and more worked on and linked to the review of the existing scientific literature, so that a work can be published in our journal,
The bibliography of the previous studies of the introduction must be connected with the discussion, that will give more power to the manuscript,
Finally, the bibliography should be ordered, following the rules of the journal.
With these changes made, the article will improve and will have the quality to be published in this prestigious magazine,
kind regards
Author Response
Ref: [Medicina] Manuscript ID: medicina-1770478 17 July 2022
To Reviewer #1
Dear Reviewer,
Thank you for your thorough analysis of our article, its overall positive evaluation, constructive comments and suggestions. We explained point-by-point the details of the revisions in the manuscript as described below in italic. Changes in the manuscript text are with "Track changes" function in Microsoft Word, so that changes are easily visible (see attached document). As there were many changes in the manuscript, we uploaded the final draft in pdf version without "Track changes" in text.
First of all, thank you for the possibility to review the manuscript.
It's a good job, on a very interesting topic as it is the satisfaction with life, Family Support, Structure and Affluence in adolescents of different countries, but before being published you must incorporate the following suggestions, I ask the authors to follow them one by one,
Response: Thank you for overall positive evaluation of our paper.
in the introduction some important references are missing, revise it.
Response: The introduction is based on a large number of studies of life satisfaction. This aspect has been central to more than 90 published scientific papers using the data collected in the Health Behaviour in School-aged Children (HBSC). Moreover, the World Happiness Report also uses the same scale for assessing happiness. In a paper of our type it is not feasible to include all relevant studies. However, we have added reference to the theoretical perspective of Diener et al. in the following revision:
"Studies show that happiness emphasizes the emotional experience, while life satisfaction includes also a cognitive component (Diener et al. 1999). Myers and Diener (1995: 11) study happiness and life satisfaction as components of subjective wellbeing. Subjective wellbeing is defined by these authors as a presence of a positive affect, absence of a negative affect and life satisfaction, i.e. the subjective wellbeing is life satisfaction, frequently experienced positive emotions (joy, affectivity) and rarely experienced negative emotions (sorrow and anxiety). Many research consider these three elements synonymous (e.g. De Neve & Oswald 2012; Veenhoven, 1996). Research shows that life satisfaction can be viewed as an important strength that facilitates young people’s development (Antaramian,2008)."
although it is deduced from the text, it must explain the research gap that the article tries to cover,
Response: To explain the research gap that the article tries to cover, the corresponding pararagraph was re-written as follows: "Research shows that family/parental support also is linked to young people's life satisfaction and well-being [52]. However, studies on the relationship between life satisfaction and family support during adolescence are relatively recent and there are still gaps in the literature in this respect [36,45]. A possible explanation for the delay in the development of research in adolescent life satisfaction study area could be related to the lack of consensus about the operationalization of the family support construct and its conceptual relationships with other family variables [37]. "
the explanation of the procedure carried out for the investigation needs to be improved, it is unclear,
Response: In the method description an overview of the procedures has been given and we also reference to the available study protocol for further information. We regard the present information is sufficient for the majority of the readers.
the discussion section should be improved and more worked on and linked to the review of the existing scientific literature, so that a work can be published in our journal,
Response: Thank you for this comment, which has resulted in a strengthening of the discussion. We have tried as much as possible to discuss the results of the study in the light of the literature review, especially the results from testing of hypotheses raised. However, our efforts were limited by the wide range of issues covered and we did not want to expand the scope of the article further, as it is large. Therefore, we only made a few additions like the following:
"In line with previous research [40–50], this study showed that perceived family support is related to adolescent life satisfaction. Moreover, this relation is intermediated with family structure and family affluence."
The bibliography of the previous studies of the introduction must be connected with the discussion, that will give more power to the manuscript,
Response: The discussion has been revised with regard to this comment as well as with regard to the above comment..
Finally, the bibliography should be ordered, following the rules of the journal.
Response: The reference list has now been organized according to the Reference List and Citations Style Guide for MDPI Journals (https://mdpi-res.com/data/mdpi_references_guide_v5.pdf).
With these changes made, the article will improve and will have the quality to be published in this prestigious magazine,
kind regards

Reviewer 2 Report
A manuscript entitled Pathways of Adolescent Life Satisfaction Association with Family Support, Structure and Affluence: A Cross Nationally Comparative Analysis examined the relationship of adolescent life satisfaction and certain family characteristics - perceived family support, family structure and family affluence, in adolescents from 45 countries.
Using a large and culturally diverse sample of adolescents, and a path analysis as a method of data analysis, the research showed that family support directly predicted life satisfaction, while family structure and family affluence both directly and indirectly (via family support, and family structure also via family affluence) predicted adolescent life satisfaction. Cross national comparisons provided interesting insights on differences in studied relationships between Eastern and Southern European countries on one hand, and Western and Central European countries on the other hand.
The study has an interesting and relevant subject for those interested in developmental psychology, adolescent development, well-being, and family as a context of development. The manuscript has a clear structure; Abstract describes the content of the paper well; the goal of the paper is clearly stated, and the study uses adequate methodology for answering its research questions. The manuscript’s results are reproducible based on the details given in the methods section (pointing the reader to the sources of more detailed information about the study). Majority of tables and figures are necessary and appropriate. Discussion and conclusions are consistent with the findings. Ethics and data availability statements are adequate.
However, I believe some major revisions are required before it can be considered for publication:
1. Please more carefully edit and proofread the entire manuscript.
2. Introduction: please consider providing some theoretical foundation for the study, not only a reference to previous literature.
3. Results: Reconsider some of the descriptive analyses (e.g. factorial ANOVAs instead of t-tests and one-way ANOVAs); Path analysis - full and partial mediation models should be compared; analyses presented in Table 8 justified.
Other specific comments and suggestions for the authors have been added directly to the text of the paper.

Author Response
Ref: [Medicina] Manuscript ID: medicina-1770478 17 July 2022
To Reviewer #2
Dear Reviewer,
Thank you for your thorough analysis of our article, its overall positive evaluation, constructive comments and suggestions. We explained point-by-point the details of the revisions in the manuscript as described below in italic. Changes in the manuscript text are with "Track changes" function in Microsoft Word, so that changes are easily visible (see attached document). As there were many changes in the manuscript, we uploaded the final draft in pdf version without "Track changes" in text.
A manuscript entitled Pathways of Adolescent Life Satisfaction Association with Family Support, Structure and Affluence: A Cross Nationally Comparative Analysis examined the relationship of adolescent life satisfaction and certain family characteristics - perceived family support, family structure and family affluence, in adolescents from 45 countries.
Using a large and culturally diverse sample of adolescents, and a path analysis as a method of data analysis, the research showed that family support directly predicted life satisfaction, while family structure and family affluence both directly and indirectly (via family support, and family structure also via family affluence) predicted adolescent life satisfaction. Cross national comparisons provided interesting insights on differences in studied relationships between Eastern and Southern European countries on one hand, and Western and Central European countries on the other hand.
The study has an interesting and relevant subject for those interested in developmental psychology, adolescent development, well-being, and family as a context of development. The manuscript has a clear structure; Abstract describes the content of the paper well; the goal of the paper is clearly stated, and the study uses adequate methodology for answering its research questions. The manuscript’s results are reproducible based on the details given in the methods section (pointing the reader to the sources of more detailed information about the study). Majority of tables and figures are necessary and appropriate. Discussion and conclusions are consistent with the findings. Ethics and data availability statements are adequate.
Response: Thank you for overall positive evaluation of our paper.
However, I believe some major revisions are required before it can be considered for publication:
- Please more carefully edit and proofread the entire manuscript.
Response: We made many revisions to the manuscript, both in terms of content, style, and language.
- Introduction: please consider providing some theoretical foundation for the study, not only a reference to previous literature.
Response: Our starting point is that life satisfaction is an assessment of overall well-being and a key predictor of many life outcomes. We give an overview of the concepts by adding:
"Studies show that happiness emphasizes the emotional experience, while life satisfaction includes also a cognitive component (Diener et al. 1999). Myers and Diener (1995: 11) study happiness and life satisfaction as components of subjective wellbeing. Subjective wellbeing is defined by these authors as a presence of a positive affect, absence of a negative affect and life satisfaction, i.e. the subjective wellbeing is life satisfaction, frequently experienced positive emotions (joy, affectivity) and rarely experienced negative emotions (sorrow and anxiety). Many research consider these three elements synonymous (e.g. De Neve & Oswald 2012; Veenhoven, 1996). Research shows that life satisfaction can be viewed as an important strength that facilitates young people’s development (Antaramian,2008)."
- Results: Reconsider some of the descriptive analyses (e.g. factorial ANOVAs instead of t-tests and one-way ANOVAs); Path analysis - full and partial mediation models should be compared; analyses presented in Table 8 justified.
Response: In regard to this comment the corrections made to the article are detailed in responses to reviewer’s questions and comments which were presented in the text of the manuscript (see below).
Other specific comments and suggestions for the authors have been added directly to the text of the paper.
Responses to comments which have been added directly to the text of the paper
PAGE 1
no need to add this information, as the authors already stated that they used path analysis
Response: A correction has been made to address this comment. The phrase ‘using the structural modelling’ was deleted.
if the authors report the results on age and gender differences in the abstract, than one would expect these differences to be one of the objectives of the study; age and gender should be mentioned previously
Response: A correction has been made to address this comment. The text ‘On average, the mean of life satisfaction score did not differ between boys and girls but decreased by age comparing 11- and 15-year olds.’ was deleted.
association of life satisfaction with physical attractiveness should not be interpreted this way;
more appropriate would be to say that those who consider themselves to be more physicaly attractive report having higher life satisfaction
Response: Text revised as follows:
"It was also demonstrated that the adolescents who consider themselves to be more physically attractive report higher life satisfaction [14], they feel also more socially competent [15], have more friends in the social media [16], have higher self-esteem [17], and experience more positive emotions [18]."
PAGE 2
not really, because happines is the affective component of the subjective well-being;
please see Ed Diener's research for subjective well-being definition
Response: We have added Ed Diener’s perspective, see above (response to comment 2).
unclear sentence; level of analysis is not relevant in this context where the authors write about determinants of adolescent life satisfaction;
Response: We corrected the corresponding sentence as follows:
"The impact of social and material resources on life satisfaction of adolescents have been in focus of several research both national [27–29] and cross-national studies [26,30–35]."
in fact, this whole paragraph should be rewised - if the authors wish to write about determinants of adolescent life satisfaction, they should do it more systematically - here they mention cross-national variations, age and gender, but none of these determinants is systematically described - how are they related to adolescent life satisfaction and why (theoretical background explaining mentioned decrease by age and gender differences)
Response: Since the first paragraph of the Introduction describes the meaning of life satisfaction for the adolescent's existence, this paragraph is intended to identify the factors on which life satisfaction depends. Indeed, this paragraph only lists these factors without going into the interaction mechanisms (otherwise it would increase the volume of the Introduction). However, the interaction mechanisms, such as the relationship of age and gender with life satisfaction, are discussed in detail in Discussion section in the context of the results obtained in the study. Taking into account the reviewer's comment, this paragraph has been revised and a consistent list of life satisfaction factors has been presented as follows:
"On the other hand, determinants of adolescent life satisfaction were also studied and many of them have been identified. The results from the Health Behaviour in School-aged Children (HBSC) surveys showed significant cross-national variation in the level of life satisfaction among adolescents [24,25]. However, despite the diversity of results, it has been revealed that life satisfaction decreases significantly during transitions in adolescence from 11-year-olds to 15-year-olds and is lower among girls [26]. The impact of social and material resources on life satisfaction of adolescents have been in focus of several research both national [27–29] and cross-national studies [26,30–35]. It has been shown that adolescent life satisfaction is strongly influenced by life experiences and relationships [17,36–39].Numerous studies also indicate that adolescent life satisfaction is associated with a range of different family characteristics [27,37,40–45]."
this sentence is almost equal in its meaning to the last sentence of the previous paragraph
Response: This sentence was removed.
not necessarily parents
Response: A correction has been made to address this comment. The phrase ‘and/or other persons’ was added.
not generally, but in a certain study the authors refer to, so this should be more clearly stated
Response: Here, the phrase ‘most important’ was changed to ‘very important’. Unfortunately, in this study, only the family affluence factor was included in the model.
please consider not using this word for non-intact families; maybe "broken family structure" would be more appropriate
Response: Here, the phrase ‘damaged’ was changed to ‘non-intact’. In the paper, ‘broken family structure’ was also used as a synonym.
how?
Response: An example was presented:
"e.g. children who have experienced parental separation may suffer from emotional distress and loss of regular contact with the non-residential parent [41,43,46,51]."
please elaborate
Response: The next paragraph explains inconsistency of associations between family factors and life satisfaction:
Research shows that family/parental support also is linked to young people's life satisfaction and well-being [52]. However, studies on the relationship between life satisfaction and family support during adolescence are relatively recent and there are still gaps in the literature in this respect [36,45]. A possible explanation for the delay in the development of research in adolescent life satisfaction study area could be related to the lack of consensus about the operationalization of the family support construct and its conceptual relationships with other family variables [37].
responsiveness or responsivity, not responsibility
Response: A correction has been made to address this comment. The phrase ‘responsiveness’ was chosen.
please elaborate - which transition?
Response: The corresponding sentence was elaborated as follows:
"However, the role of parents and parent-adolescent relations undergo a process of change through the maturation of the child during adolescence and subsequent transition from childhood to adulthood."
PAGE 3
why would this be "DESPITE the evidence of developmental changes"?
Response: We deleted ‘despite the evidence of developmental changes in perceived family support’.
PAGE 4
please provide information about the study subjects - sample size, percentage of boys and girls in the sample
Response: Only the following information has been provided here:
"After combining data from 45 countries, it included 228,979 records. The present analysis used records of 188,619 students who reported all variables of the model structure."
Information about percentage of boys and girls in the sample is provided in Table 1. Since gender is one of the factors used in the model, we believe that information about this factor should be provided in the Results section.
PAGE 5
this information is relevant in the next section, not in this one
Response: Here we just wanted to point out that the adolescent's gender and age were also recorded. This information has been revised and left in the same place.
"Gender and age (11-, 13-, and 15-year-old) of the adolescent were also recorded during data collection."
this sentence is not clear - comparison of which means? relationship between which variables?and, most importantly, how is the comparison of the means related to PREDICTION of strength and direction of the relationship between variables?
which two groups?
which three groups?
Response: In light of these comments, the relevant text has been amended to read as follows:
"Comparison of the means of life satisfaction score between groups of adolescents was performed using one-way ANOVA test, while Bonferroni test was performed as a post-hoc analysis for comparison of the means between three groups of adolescents. This analysis made it possible to predict the strength of the relationship between life satisfaction and other variables and the "direction" (positive or negative) of the relationship."
references not needed
Response: We consider it needs for readers who are little familiar with the path analysis.
this information should be reported earlier, when describing the sample
Response: Since gender is one of the factors used in the model, we believe that information about this factor should be provided in the Results section.
PAGE 8
why t-tests and one-way ANOVA, why not factorial 2x3 ANOVA? that would be a more appropriate analysis
factorial ANOVAs would be more appropriate analyses for these data also;
Response: We appreciate this insightful observation and in this study statistical significance was tested by ANOVA test. We will also use it in the future research.
please have in mind that analyses should be made following research problems; I understand that this part is what the authors refer to as "descriptive results" but it should be clear what they wish to present here and why;
correlation analysis is what usually precedes path analysis
Response: The result section has been divided into three sub-sections, which gives a more concise presentation of the results. The first subsection is an introduction to the modeling. We agree with the reviewer's opinion that pairwise correlational analysis is usually chosen for this purpose. This applies when both variables are of continuous type. Unfortunately, in this study we examined the relationships when one of the two variables was categorical, so in this case the relationship between the variables can be assessed by comparing the mean of the continuous variable in groups of subjects defined according to the second (categorical) variable. Thus, this analysis made it possible to determine the strength of the relationship between the variables and the "direction" of the relationship (is it positive or negative).
PAGE 9
the hypothesized model should be presented graphically (by path diagram) before its goodness-of-fit indices are shown and analysed
Response: The table of goodness-of-fit indices was moved to the end subsection 3.2.
I am not sure there are more and less important fit indices, they all tell us something about the degree of data-model fit, but none of them is "the best" or "most important"
Response: We decided to change the corresponding statement as follows:
"The χ2/(df) statistic that was applied to assess the magnitude of the discrepancy between the sample and fitted covariance matrice indicated consistency (p >0.05) the model and data only in a few countries due to its sensitivity to sample size. The RMSEA index showed ... "
when previously describing fit indices that will be used, the authors did not mention chi square/df ratio; please add this in your section 2.4. statistical analysis
Response: A correction has been made to address this comment.
PAGE 10
A standardized regression coefficient of 0.103 absolute value would not be considered strong
Response: We agree it is not high. A correction has been made to address this comment by changing the phrase ‘strong’ to ‘statistically significant’.
for a more systematic approach to testing the hypothesized model, the authors should compare partial vs. full mediation models by trimming direct paths in the model and comparing the relative fit of nested models
Response: This step of analysis was performed and described previously (see description of the Figure 1):
"Considering the effects of gender, age, family structure, and family affluence on family support, the effect of the latter on life satisfaction was reduced by a total of 0.07 units. Thus, the direct effect of family support on life satisfaction can be predicted to be at the level of 0.49, which was confirmed by a separate path model without indirect effects (data are not presented)."
PAGE 11
this was not a variable in the analysis (?)
Response: ‘performance’ was changed to ‘life satisfaction’.
in my opinion, there is no need for this Estonian example, graphically or in the text, but there is a need to show this model as the hypothesized one, as I mentioned earlier in my comments
Response: We considered it important to show the details of the analysis in one country. The results of the path analysis in all 45 countries are summarized in Table 6 and in the text.
PAGE 12
I have never come across these analyses, I am not sure that it is meaningful or justified to calculate correlations of path coefficients and R2;
could authors please provide a rationale for these analyses, and a reference for it?
Response: I regard to this comment several essential corrections have been completed. First, in Results, a new subsection titled as “3.3. Groups of Countries According to the Characteristics of the Path Model“ was formed. It also includes a new paragraph where selection of variables for cluster analysis is described. Description of the Table 8 was removed to the new 3.3 subsection.
This correction is:
"3.3. Groups of Countries According to the Characteristics of the Path Model
As 45 countries were involved in the current round of the HBSC study, there were the same number of different path models constructed. Consequently, we tried to group the countries according to the main characteristics of the path model. Some of these model characteristics (R2, LS mean, β1 (LS ß FS)) were selected using the results of correlation analysis, which evaluated the relationships between R2, mean of life satisfaction score and path regression weights estimated from the country level data (Table 8). The analysis revealed that R2 had the maximal correlation (r=0.935, p<0.01) with the regression weight of family support on life satisfaction score but was uncorrelated with the regression weights of age (r=0.076, p=0.622) and family affluence (r=–0.187, p=0.220) on life satisfaction score. Meanwhile, higher R2 values were observed in countries with lower mean of life satisfaction score (r=–0.591, p<0.01). Additionally, the strength of life satisfaction association with age (β4 (LS ß AGE)) was chosen for classification of the models because it, as well as the strength of life satisfaction association with family support (β1 (LS ß FS)), was significant in all countries.
Second, the Discussion section was supplemented by a rationale for use of selected model characteristics in relation to the considered model. The rationale to calculate correlations between path coefficients and R2 was not discussed. This correction is:
"In the presented model, R2 had a strong positive relationship with the strength of association between family support and life satisfaction, but was negatively associated with the level of life satisfaction. The calculated correlations allowed to identify the conditions on which the quality of the model depends.
This study is unique as a series of path models was constructed using data from a variety of countries. Despite that each model had unique characteristics, some similarities within several model groups could be seen, so cluster analysis was used to classify the countries according to the model characteristics. For this purpose, we selected four model characteristics which were considered consistent from a statistical point of view. These were R2, mean level of life satisfaction, the strength of life satisfaction association with family support, and the strength of life satisfaction association with age. According to these characteristics..."
At the same time, we would like to thank reviewer for asking the question whether it is possible to calculate the correlations presented in this article. Unfortunately, nothing is written about this kind of analysis in the statistical literature because there are very few multicenter data, such as those used in this study. The statistics calculated in this paper are based on country-level data from a sufficiently large sample (45 countries) to be statistically significant. In our opinion, the calculated correlations make a lot of sense because they allow us to identify the conditions on which the quality of the model depends.
